# Gastroesophageal reflux symptoms and sleep quality among medical students at a private university in Lima, Peru: A cross-sectional study

**Alvaro F. Montalvo-Peralta, Yolanda Abigail Miranda-Sagastegui, Frank J. Tinco-Pumacahua, David R. Soriano-Moreno**[iD]*

School of Medicine, Universidad Peruana Unión, Lima, Peru

* davidsoriano@upeu.edu.pe

## Abstract

### Introduction

Poor sleep quality is common among medical students and is associated with academic and health-related consequences. Gastroesophageal reflux disease (GERD) may contribute to sleep disturbances through biological and behavioral mechanisms, however, evidence in medical students is limited and previous studies have not adequately controlled for confounding.

### Objective

To evaluate the association between GERD and sleep quality among medical students in Lima, Peru.

### Methods

Cross-sectional analytical study conducted among medical students from a private university in Lima, Peru. Sleep quality was assessed using the Pittsburgh Sleep Quality Index, and GERD symptoms were measured with the Frequency Scale for the Symptoms of GERD. Poisson regression with robust variance was used to estimate adjusted prevalence ratios (aPR) with 95% confidence intervals (95% CI).

### Results

A total of 171 students were analyzed (median age: 22 years; 64.9% female). The prevalence of GERD symptom burden was 76.6%, and poor sleep quality was observed in 84.8% of participants. Students with GERD symptom burden had a higher prevalence of poor sleep quality compared to those without GERD (aPR: 1.27; 95% CI: 1.03 to 1.58). Additionally, for each one-point increase in the GERD symptom score, the prevalence of poor sleep quality increased by 1.0% (aPR: 1.01; 95% CI: 1.003 to 1.018), demonstrating a dose–response relationship.

**Data availability statement:** Yes - all data are fully available without restriction; All relevant data are within the paper and its Supporting Information files.

**Funding:** The author(s) received no specific funding for this work.

**Competing interests:** The authors have declared that no competing interests exist.

## Conclusion

GERD symptoms were significantly associated with poor sleep quality among Peruvian medical students. This study provides context-specific adjusted evidence from an under-studied Latin American medical student population. These findings support the need for screening strategies and early interventions targeting both conditions in this population.

## Introduction

Sleep quality is defined as an individual's self-satisfaction with all aspects related to the sleep experience [1]. The prevalence of poor sleep quality in adults can reach up to 32.8% [2], affecting medical students more frequently, where prevalence has been reported to reach 55.6%, with an estimated average of 6.5 hours of sleep per night [3]. Sleep-related problems have been associated with increased stress [4], depression [5], poorer physical health [6], and worse academic performance [7].

Previously, several factors associated with poor sleep quality among medical students in Latin America have been identified, including female sex, tobacco use, and the presence of depressive and anxiety symptoms [8]. However, the influence of digestive disorders is less clear. Gastroesophageal reflux disease (GERD) is common, with a prevalence close to 14%, higher in Latin America and possibly even higher among medical students [9–11]. Biologically, esophageal acid exposure may activate receptors such as TRPV1, stimulating alertness and pain mechanisms that interfere with sleep [12]; in turn, sleep deprivation increases esophageal hypersensitivity and acid exposure time [13]. Alterations in melatonin and inflammatory processes may further reinforce this bidirectional relationship [14].

A systematic review found that GERD is significantly associated with poorer sleep quality (OR: 1.47; 95% CI: 1.24 to 1.74) and shorter sleep duration (OR: 1.17; 95% CI: 1.12 to 1.21), although with some heterogeneity among studies [14]. Evidence in medical students is still scarce. A study in Iran reported an association between reflux symptoms and impaired sleep quality, while a study in India found that inadequate sleep and sleep-related habits were associated with GERD symptoms; however, both studies were cross-sectional and lacked adequate adjusted analyses to control for confounding [15,16]. This is particularly important in medical students, a population with a high frequency of poor sleep quality, irregular eating habits, and heavy academic workload, factors that may modify this relationship. To date, no study with adequate control for confounding has evaluated this association among Peruvian medical students. Therefore, context-specific evidence is needed to better characterize this relationship and support local preventive strategies.

Therefore, this study aimed to evaluate the association between GERD symptoms and sleep quality among medical students at a private university in Lima, Peru.

## Methods

### Design, population and sample

An analytical cross-sectional study was conducted among medical students from a private university (Universidad Peruana Unión) in Lima, Peru, between October

27 and November 28, 2024. This university has a campus in Lima where the study was carried out within the Faculty of Health Sciences, specifically in the School of Medicine. The medical program is a face-to-face undergraduate program with a duration of seven academic years, structured into three training stages: basic sciences, clinical sciences, and internship. We included medical students aged ≥18 years, who voluntarily agreed to participate in the study. Students with incomplete data were excluded.

A sample size calculation specifically for detecting an association was not performed because no preliminary evidence was available in this population using the same instruments to assess both GERD symptoms and sleep quality. Therefore, the sample size was estimated using a descriptive approach in Epidat V4.2 software, considering a finite population of 698 students, a proportion of poor sleep quality of 83.9% [17], a 95% confidence level, and a 5% margin of error, resulting in a minimum required sample size of 160 participants. Non-probabilistic convenience sampling was used because no official student list was available to perform random sampling.

## Variables

The dependent variable was sleep quality during the last month, measured using the Pittsburgh Sleep Quality Index (PSQI), which includes 19 items grouped into seven sleep components: subjective sleep quality, sleep latency, sleep duration, habitual sleep efficiency, sleep disturbances, use of sleep medication, and daytime dysfunction. This questionnaire has been previously validated in Peruvian university students and shows adequate reliability (Cronbach's alpha = 0.79) [18]. The global score ranges from 0 to 21, with higher scores indicating worse sleep quality. Poor sleep quality was defined as a score ≥5, and good sleep quality as a score <5 [19,20].

The independent variable was GERD symptoms, measured using the Frequency Scale for the Symptoms of Gastroesophageal Reflux Disease (FSSG), which has been used previously among Peruvian medical students [21,22]. This instrument includes 12 questions assessing reflux and dyspeptic symptoms using a Likert-type scale scored as 0 "never", 1 "occasionally", 2 "sometimes", 3 "often", and 4 "always". The total score ranges from 0 to 48 points, with higher scores indicating greater presence of GERD symptoms. GERD symptom burden was defined as a total score ≥8 points.

Other variables evaluated included sociodemographic data such as age (years), sex, academic year, body mass index, and current employment status. Additionally, consumption habits of nonsteroidal anti-inflammatory drugs (NSAIDs), energy drinks, spicy foods, coffee, alcoholic beverages, and tobacco were assessed.

## Procedures

Prior to data collection, approval from the ethics committee and authorization from the medical school were obtained. An online survey was designed using Microsoft Forms. The survey was distributed during an in-person academic event organized by the medical school, which included students from all academic years. This event is usually mandatory and forms part of the academic evaluation, which likely reduced the possibility of substantial selection bias due to absenteeism. In that context, attendees were invited to voluntarily complete the online form through a link provided by the research team. Additionally, the survey link was disseminated via WhatsApp to increase participation. Data collection was carried out between October 27 and November 28, 2024. The survey was structured into sections in the following order: (1) informed consent, (2) clinical and demographic characteristics, (3) GERD symptom questionnaire (FSSG), and (4) sleep quality questionnaire (PSQI).

## Statistical analysis

Data from the online survey were exported to Microsoft Excel, where data cleaning and coding were performed. During this process, the database was reviewed to identify duplicate records, missing values, and implausible entries, but no such issues were found. Subsequently, the statistical software STATA V19.0 was used for analysis.

 

Categorical variables were described as absolute and relative frequencies, and numerical variables as mean with standard deviation or median with interquartile range depending on their distribution. For the bivariate analysis according to sleep quality, Chi-square and Fisher's exact tests were used for categorical variables, and the Mann–Whitney U test for numerical variables. The association between GERD and sleep quality was evaluated using Poisson regression with robust variance to calculate crude prevalence ratios (cPR) and adjusted prevalence ratios (aPR). This regression approach was used given the high-prevalence binary nature of the outcome. Robust (sandwich) variance estimators were applied to obtain valid standard errors when Poisson models are fitted to binary data. The association was assessed using the GERD symptom score (continuous) and the presence or absence of GERD symptom burden according to the cutoff of ≥8 points (dichotomous). For inclusion of variables in the adjusted model, we followed an epidemiological approach by including all potential confounders of the association based on clinical relevance, biological plausibility, and prior literature, rather than on statistical significance in the bivariate analysis. Accordingly, all sociodemographic and lifestyle variables considered plausible confounders of the association were included in the adjusted model regardless of their bivariate p-values [23,24]. Multicollinearity was assessed using the variance inflation factor (VIF). We observed that age, when included as a continuous variable, showed a high degree of multicollinearity. To reduce this collinearity while retaining age as a covariate in the adjusted model, age was categorized into tertiles. A graph showing the relationship between the GERD symptom score and the adjusted prevalence of sleep quality was generated using post-estimation commands from the adjusted model. A p-value <0.05 was considered statistically significant.

### Ethical considerations

The protocol of this study was approved by the Ethics Committee of Universidad Peruana Unión (approval code: 2024-CEB-FCS-UPeU-N°197) and was conducted in accordance with institutional ethical principles and the Declaration of Helsinki. Participation was voluntary, and informed consent was obtained prior to completion of the online survey. The collected information was handled confidentially.

## Results

Initially, 176 students were surveyed, of whom 5 participants aged under 18 years were excluded; therefore, 171 participants were finally analyzed (Fig 1).

The included students had a median age of 22 years [IQR: 20–24], with the majority being female (64.9%) and in the first year of the program (21.6%). Regarding consumption habits, the use of NSAIDs (36.3%), energy drinks (31.0%), coffee (43.9%), and spicy foods (49.7%) were common. In contrast, alcohol (17.0%) and tobacco (5.9%) consumption were less prevalent. Regarding GERD symptoms, the mean score of the scale was 13.6±8.4, with a GERD symptom burden prevalence of 76.6% (Table 1). Detailed GERD symptoms are presented in S1 Table.

Regarding sleep quality, the mean PSQI score was 7.8±3.3, with sleep duration being the most frequently affected domain (S2 Table). The prevalence of poor sleep quality was 84.8%, being more frequent among those with higher GERD symptom scores (mean 14.5±8.4, p<0.001) and among those with GERD (90.1%, p=0.001). Additionally, prevalence was significantly higher among students who reported NSAID consumption (93.5%, p=0.016), energy drink consumption (94.3%, p=0.012), and spicy food consumption (91.8%, p=0.012) (Table 2).

In the adjusted regression analysis, students with GERD symptom burden had a 27% higher prevalence of poor sleep quality compared with those without GERD (aPR: 1.27; 95% CI: 1.03 to 1.58; p=0.028). Likewise, for each additional point in the GERD symptom score, the prevalence of poor sleep quality increased by 1.0% (aPR: 1.010; 95% CI: 1.003 to 1.018; p=0.005) (Table 3). Graphically, a positive association was observed between increasing GERD symptom score and the adjusted prevalence of poor sleep quality (Fig 2).

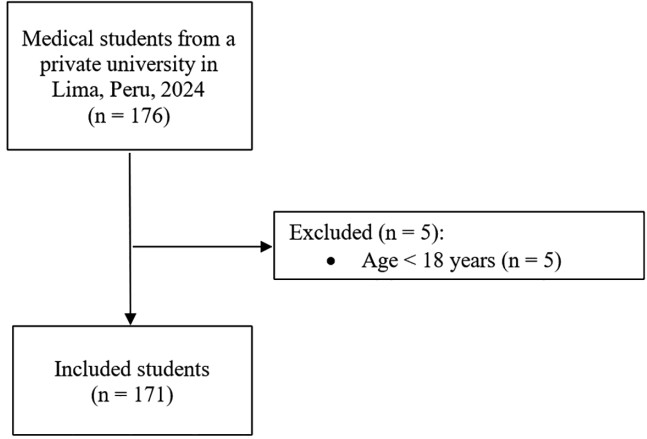

**Fig 1. Flowchart of participant selection.**

## Discussion

### Main findings

We found that medical students showed a high prevalence of GERD symptom burden (76.6%) and poor sleep quality (84.8%). Additionally, the presence of GERD symptom burden or higher GERD symptom scores were associated with higher prevalences of poor sleep quality.

Previous studies among Peruvian university students have reported similar or even lower prevalences of GERD symptom burden. A research conducted among medical students at another Peruvian university reported a GERD prevalence of 75%, while another study conducted in medical students in Huacho, Peru, reported a lower prevalence of 42% [22,25]. However, the frequency observed in our study greatly exceeds the global estimate (14%) and that reported for Latin America (12.9%) [9]. Several factors may explain these differences. The use of the FSSG captures the frequency of reflux-related symptoms rather than a clinically confirmed diagnosis, which may yield higher prevalence estimates. Also, convenience sampling may have favored the participation of students who were more health-conscious or more likely to report symptoms. Furthermore, the high frequency of unhealthy dietary and behavioral habits identified in the sample, such as excessive consumption of coffee and energy drinks, use of NSAIDs, and intake of spicy foods, may have contributed to the elevated prevalence observed [26,27]. Additional context-specific factors that could explain the elevated GERD prevalence include academic stress, sleep deprivation, irregular eating schedules, and greater awareness and reporting of symptoms due to medical training.

Another important finding was the high prevalence of poor sleep quality. This figure exceeds the estimate reported in a global systematic review that included more than 50,000 medical students and reported a prevalence of 55.6% [3]. On the other hand, this prevalence is consistent with results from previous studies conducted among Peruvian medical students that used the PSQI scale [17,28]. This high frequency could be attributed to the intense academic workload, sustained stress, high prevalence of depression and anxiety, irregular schedules for studying and resting, and frequent use of electronic devices at night, factors that disrupt circadian rhythms and compromise both sleep duration and quality [8,29]. In addition, the competitive environment and the academic demands typical of medical schools in the Peruvian context may contribute to persistent patterns of sleep restriction and fragmentation [30].

Regarding the association between GERD and sleep quality, a previous systematic review including 22 case–control or cohort studies found that GERD was associated with higher odds of poor sleep quality (OR: 1.47; 95% CI: 1.21 to 1.79)

**Table 1. Characteristics of medical students from a private university in Lima, Peru (n = 171).**

| Characteristics | n (%)* |
|---|---|
| Sex | |
| Male | 60 (35.1) |
| Female | 111 (64.9) |
| Age (years), median [IQR] | 22 [20 –24] |
| Year of study | |
| First year | 37 (21.6) |
| Second year | 20 (11.7) |
| Third year | 18 (10.5) |
| Fourth year | 27 (15.8) |
| Fifth year | 21 (12.3) |
| Sixth year | 35 (20.5) |
| Seventh year | 13 (7.6) |
| Body mass index | |
| Underweight | 5 (2.9) |
| Normal | 106 (62.0) |
| Overweight | 50 (29.2) |
| Obesity | 10 (5.9) |
| Currently working | |
| No | 164 (95.9) |
| Yes | 7 (4.1) |
| NSAID use | |
| No | 109 (63.7) |
| Yes | 62 (36.3) |
| Energy drink consumption | |
| No | 118 (69.0) |
| Yes | 53 (31.0) |
| Spicy food consumption | |
| No | 86 (50.3) |
| Yes | 85 (49.7) |
| Coffee consumption | |
| No | 96 (56.1) |
| Yes | 75 (43.9) |
| Alcohol consumption | |
| No | 142 (83.0) |
| Yes | 29 (17.0) |
| Tobacco consumption | |
| No | 161 (94.2) |
| Yes | 10 (5.9) |
| Total GERD symptom scale score, mean ± SD | 13.6 ± 8.4 |
| GERD symptom burden according to score (≥ 8) | |
| No GERD | 40 (23.4) |
| GERD | 131 (76.6) |
| Total Pittsburgh score (PSQI), mean ± SD | 7.8 ± 3.3 |
| Sleep quality according to PSQI (≥ 5) | |
| Good sleep quality | 26 (15.2) |
| Poor sleep quality | 145 (84.8) |

*(Continued)*

**Table 1.** (Continued)

GERD: gastroesophageal reflux disease; PSQI: Pittsburgh Sleep Quality Index; NSAIDs: nonsteroidal anti-inflammatory drugs; SD: standard deviation; IQR: interquartile range.

* Absolute frequencies (n) and relative frequencies (%)

**Table 2.** Bivariate analysis of sample characteristics according to sleep quality among medical students from a private university in Lima, Peru (n = 171).

| Characteristics | Sleep quality (PSQI) | | p value |
| --- | --- | --- | --- |
| | Good 26 (15.2%) n (%) | Poor 145 (84.8%) n (%) | |
| Sex | | | 0.695[a] |
| Male | 10 (16.7) | 50 (83.3) | |
| Female | 16 (14.4) | 95 (85.6) | |
| Age (years), median [IQR] | 21 [19 –23] | 22 [20 –24] | 0.388[b] |
| Year of study | | | 0.763[a] |
| Basic sciences (1st to 2nd year) | 8 (14.0) | 49 (86.0) | |
| Clinical sciences (3rd to 7th year) | 18 (15.8) | 96 (84.2) | |
| Body mass index | | | 0.753[a] |
| Normal/ underweight | 16 (14.4) | 95 (85.6) | |
| Overweight | 9 (18.0) | 41 (82.0) | |
| Obesity | 1 (10.0) | 9 (90.0) | |
| NSAID use | | | 0.016[a] |
| No | 22 (20.2) | 87 (79.8) | |
| Yes | 4 (6.5) | 58 (93.5) | |
| Energy drink consumption | | | 0.020[a] |
| No | 23 (19.5) | 95 (80.5) | |
| Yes | 3 (5.7) | 50 (94.3) | |
| Spicy food consumption | | | 0.012[a] |
| No | 19 (22.1) | 67 (77.9) | |
| Yes | 7 (8.2) | 78 (91.8) | |
| Coffee consumption | | | 0.059[a] |
| No | 19 (19.8) | 77 (80.2) | |
| Yes | 7 (9.3) | 68 (90.7) | |
| Alcohol consumption | | | 0.053[c] |
| No | 25 (17.6) | 117 (82.4) | |
| Yes | 1 (3.4) | 28 (96.6) | |
| Tobacco consumption | | | 0.363[c] |
| No | 26 (16.1) | 135 (83.9) | |
| Yes | 0 (0.0) | 10 (100.0) | |
| Total GERD score, mean ± SD | 8.5 ± 6.0 | 14.5 ± 8.4 | <0.001[b] |
| GERD symptom burden according to score (≥ 8) | | | 0.001[a] |
| No GERD | 13 (32.5) | 27 (67.5) | |
| GERD | 13 (9.9) | 118 (90.1) | |

SD: standard deviation; IQR: interquartile range; PSQI: Pittsburgh Sleep Quality Index; GERD: gastroesophageal reflux disease; NSAIDs: nonsteroidal anti-inflammatory drugs.

[a]Chi-square test.

[b]Mann–Whitney U test.

[c]Fisher's exact test

**Table 3. Association between GERD symptoms and sleep quality (n = 171).**

| Variable | Crude PR (95% CI) | p value | Adjusted PR (95% CI)* | p value |
|---|---|---|---|---|
| GERD symptom burden | | | | |
| Absent | Ref. | | Ref. | |
| Present (score ≥ 8) | 1.33 (1.07 to 1.68) | 0.011 | 1.27 (1.03 to 1.58) | 0.028 |
| Total GERD symptom score | 1.013 (1.006 to 1.020) | <0.001 | 1.010 (1.003 to 1.018) | 0.005 |

PR: prevalence ratio; 95% CI: 95% confidence interval; Ref.: reference.

*Adjusted for age (tertiles), sex, body mass index, year of study, alcohol consumption, coffee consumption, tobacco use, NSAID use, energy drink consumption, and spicy food consumption

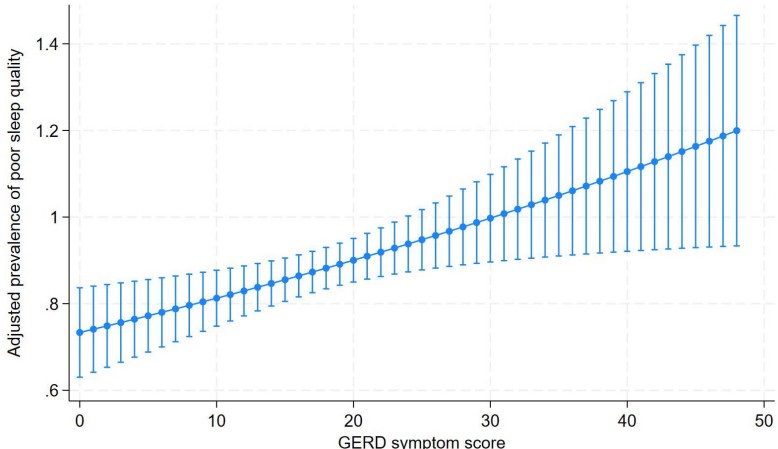

**Fig 2. Relationship between GERD symptoms and the adjusted prevalence of poor sleep quality (PSQI).**

and shorter sleep duration (OR: 1.17; 95% CI: 1.12 to 1.21) [14]. Similarly, a previous cross-sectional study conducted in Iran found that medical students with GERD symptoms had higher frequencies of poor sleep quality (p<0.001); however, adjusted regression analyses to control for confounding were not performed [16]. Another cross-sectional study conducted among medical students in India found that inadequate sleep was not associated with the presence of GERD (p=0.260); however, going to sleep within one hour after dinner was associated with a higher frequency of GERD (p=0.001) [15]. Our findings are consistent with the systematic review and the Iranian study, as we also found that GERD symptom burden was associated with a higher prevalence of poor sleep quality, but differ from the Indian study, which did not observe this association, probably due to methodological differences such as the lack of adjusted analyses and differences in the sleep-related measures used. Moreover, in our study we observed a dose–response relationship between greater GERD symptomatology and a higher prevalence of poor sleep quality, which strengthens the plausibility of the association. From a biological plausibility perspective, nocturnal reflux may induce micro-arousals and sleep fragmentation due to nocturnal acid exposure and autonomic alterations that reduce the parasympathetic activity typical of sleep [31–33]. However, given the cross-sectional design, temporality cannot be established, and the relationship may be bidirectional, as suggested by previous evidence [14].

## Implications

Our study provides several incremental contributions to the existing literature on the relationship between GERD and sleep quality. First, it focuses on Peruvian medical students, a specific and understudied subgroup within the Latin

American context. Second, unlike previous meta-analysis [14], which was based primarily on studies from the general population, our study analyzes a population with a distinct risk profile characterized by high academic stress, irregular dietary habits, and young age. Third, by using Poisson regression with robust variance, we obtained adjusted prevalence ratios that are more appropriate for cross-sectional, representing a methodological improvement over previous studies in this population that relied mainly on unadjusted comparisons. Finally, we identified a dose–response relationship between GERD symptom burden and poor sleep quality, which strengthens the biological plausibility of the observed association in this specific population.

### Recommendations

In this context, systematic screening for GERD symptoms and sleep disturbances could facilitate early detection and timely management, considering that both are potentially modifiable factors and may reinforce each other. Furthermore, interventions aimed at modifying risk behaviors such as improving diet quality, regulating meal schedules (avoiding food intake close to bedtime), indicating pharmacological treatment when clinically justified, reducing energy drink consumption, and strengthening sleep hygiene together with a comprehensive approach to common mental health problems in this population could help reduce the burden and impact of both conditions [34].

### Limitations and strengths

Some limitations should be considered. The cross-sectional design prevents establishing causality and directionality of the association. Convenience sampling may limit external validity and generate selection bias, as recruitment during an in-person academic event and through WhatsApp may have favored the participation of more health-conscious or more engaged students, potentially leading to overestimation of the prevalences of both GERD symptoms and poor sleep quality Self-reported nature of the data may have affected internal validity through reporting biases, including recall bias and social desirability bias. In addition, FSSG and PSQI are screening instruments and do not correspond to a confirmed clinical diagnosis and although the FSSG has been used in Peruvian medical students, its psychometric properties have not been formally validated in this local population, which may affect the accuracy of GERD symptom classification. Furthermore, the PSQI assesses sleep quality during the last month, whereas the FSSG does not specify an explicit recall period, which may have introduced temporal misalignment between the measurement of the exposure and the outcome. This may have introduced non-differential misclassification of the exposure, potentially biasing the observed association toward the null. Another important factor is that variables such as academic stress, mental health factors, and sleep medication use were not measured, which could have resulted in residual confounding and biased the observed association in either direction. As strengths, validated and widely used instruments were employed to measure GERD and sleep quality. In addition, adjusted regression analyses were performed including the main confounding variables, unlike previous studies conducted among medical students. The use of both dichotomous and continuous GERD measures also allowed a more comprehensive assessment of the association, including a dose–response pattern.

## Conclusion

Among Peruvian medical students, a high prevalence of both GERD symptomatology and poor sleep quality was observed. The presence of GERD symptom burden was significantly associated with a higher prevalence of poor sleep quality, and a dose–response relationship was also observed, in which higher GERD symptom scores were related to a proportional increase in the prevalence of poor sleep quality. These findings highlight the use of simple screening tools for GERD symptoms and sleep quality in university health settings and the implementation of targeted interventions focused on sleep hygiene and reduction of reflux-related behaviors. Students with persistent or more severe symptoms could also benefit from referral pathways for clinical evaluation and management.

## Supporting information

**S1 Table. Frequency of GERD symptoms (n = 171).**
(DOCX)

**S2 Table. Sleep quality according to the components of the Pittsburgh Sleep Quality Index (PSQI) (n = 171).**
(DOCX)

**S1 File. Data.**
(XLSX)

## Author contributions

**Conceptualization:** Alvaro F. Montalvo-Peralta, Yolanda Abigail Miranda-Sagastegui.

**Data curation:** David R. Soriano-Moreno.

**Formal analysis:** David R. Soriano-Moreno.

**Investigation:** Alvaro F. Montalvo-Peralta, Yolanda Abigail Miranda-Sagastegui, Frank J. Tinco-Pumacahua, David R. Soriano-Moreno.

**Methodology:** David R. Soriano-Moreno.

**Project administration:** Alvaro F. Montalvo-Peralta, Yolanda Abigail Miranda-Sagastegui, Frank J. Tinco-Pumacahua, David R. Soriano-Moreno.

**Software:** David R. Soriano-Moreno.

**Supervision:** David R. Soriano-Moreno.

**Writing – original draft:** Alvaro F. Montalvo-Peralta, Yolanda Abigail Miranda-Sagastegui, Frank J. Tinco-Pumacahua.

**Writing – review & editing:** David R. Soriano-Moreno.

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
