## [Decision Letter · Decision Letter 0]

6 Apr 2026

PONE-D-26-11670Association between gastroesophageal reflux symptoms and sleep quality among medical students at a private university in Lima, Peru: a cross-sectional studyPLOS One

Dear Dr. Soriano-Moreno,

Thank you for submitting your manuscript to PLOS ONE. After careful consideration, we feel that it has merit but does not fully meet PLOS ONE’s publication criteria as it currently stands. Therefore, we invite you to submit a revised version of the manuscript that addresses the points raised during the review process.

We look forward to receiving your revised manuscript.

Kind regards,

Taeyun Kim

Academic Editor

PLOS One

**Journal Requirements:**

6. Thank you for providing your underlying data as Supporting Information.

We note that the data set contains text or data that is not in English. Please note that PLOS is an English-language publisher, so we require data sets to be provided in English as well. Please upload an English-language version of your data set.

This will also allow us to determine if your data follows PLOS standards per our Data Availability policy here: https://journals.plos.org/plosone/s/data-availability

7. We note that there is identifying data in the Supporting Information file < Data.xlsx >. Due to the inclusion of these potentially identifying data, we have removed this file from your file inventory. Prior to sharing human research participant data, authors should consult with an ethics committee to ensure data are shared in accordance with participant consent and all applicable local laws.

-Location data

Please remove or anonymize all personal information (Age, and ID numbers), ensure that the data shared are in accordance with participant consent, and re-upload a fully anonymized data set. Please note that spreadsheet columns with personal information must be removed and not hidden as all hidden columns will appear in the published file.

Reviewers' comments:

Reviewer's Responses to Questions

**Comments to the Author**

1. Is the manuscript technically sound, and do the data support the conclusions?

Reviewer #1: Yes

Reviewer #2: Partly

Reviewer #3: Partly

Reviewer #4: Yes

2. Has the statistical analysis been performed appropriately and rigorously? 

Reviewer #1: Yes

Reviewer #2: Yes

Reviewer #3: Yes

Reviewer #4: Yes

3. Have the authors made all data underlying the findings in their manuscript fully available?

Reviewer #1: No

Reviewer #2: Yes

Reviewer #3: Yes

Reviewer #4: Yes

4. Is the manuscript presented in an intelligible fashion and written in standard English?

Reviewer #1: Yes

Reviewer #2: Yes

Reviewer #3: Yes

Reviewer #4: Yes

5. Review Comments to the Author

Reviewer #1: In the methods section, consider presenting the sample size calculation first, then the recruitment approach, and finally how incomplete records were handled.

The study does not report the number of students approached or the number who declined participation

The definition of poor sleep quality requires clarification. The manuscript classifies poor sleep as a PSQI score ≥5; however, the validated cutoff defines poor sleep quality as a global PSQI score >5, with scores ≤5 indicating good sleep quality. The authors should clarify/justify the use of this alternative threshold

The link between multicollinearity assessment (VIF) and the decision to categorize age into tertiles requires clarification. VIF is used to detect collinearity among predictors. Please clarify the methodological basis for this decision.

Given the findings of reference 14, the incremental contribution of this study is not immediately clear. The discussion should explicitly articulate how this study advances existing knowledge, including any novel population-specific insights or methodological contributions.

Reviewer #2: In this work, Alvaro F. Montalvo-Peralta and colleagues conducted a cross-sectional study to examine the association between gastroesophageal reflux symptoms and sleep quality among medical students at a private university in Lima, Peru. Their findings highlight the alarming burden of GERD and Poor sleep quality among medical students and the significant association between GERD symptoms and poor sleep quality. The data is interesting, and the topic is relevant, and I want to congratulate the authors for their work. However, I have several comments for the authors to improve the clarity of the manuscript.

General comment

• Overall, check for grammatical flow and sentence formation of the entire manuscript, as some of the sentences are long, vague, and not up to the scientific standard.

Title and Abstract

• Limit abbreviations in the abstract section.

Introduction

• The authors didn’t clearly describe what is already known regarding the association (and its degree) b/n sleep quality and GERD symptoms, particularly in the study population (medical students)?

• Which gaps in the literature does the current study aim to narrow?

Methods

• In the first paragraph of the methods section, “An analytical cross-sectional study was conducted among medical students from a private university (Universidad Peruana Unión) in Lima, Peru, during 2024”.

o The study period should be stated (eg, from October 27 to November 28, 2024)

• Inclusion and exclusion criteria should be clearly delineated and justified.

o Given that the study population is medical students (who are most likely more than or somewhere around 18 years old), what is the rationale for using the 18-year-old age cutoff for inclusion?

o Were the exclusion criteria limited to participants with incomplete data? Please mention if any additional exclusion was used? This is important for clarity and transparency.

o What defines incomplete data in the study?

• In participant selection, why do you prefer to include only students attending an in-person academic event? This can introduce a potential selection bias because students with frequent GERD symptoms and poor sleep may have increased absenteeism, which could lead to their exclusion and underrepresentation in the study.

• Why did you prefer to use non-probabilistic convenience sampling? Even though non-probabilistic convenience sampling can be useful for pilot studies or exploratory research, in analytical studies like this, it can pose a significant limitation. The authors should clarify these points and acknowledge them as limitations if necessary.

• Was block sampling by academic year used to affirm the balanced distribution of participants across academic years?

• Was sample size calculation for detecting associations performed?

• The dependent variable in the study was sleep quality, specifically “during the last month”. Do you use a similar time frame for the independent variable (GERD symptoms)? This is critical to ensure a temporal association between GERD symptoms and sleep quality

• How do you select covariates? Some relevant confounders based on prior literature seem to be missing. This includes

o Use of symptomatic treatments for GERD, like PPIs

o Comorbidities, including psychiatric illness and other medical conditions known to impact sleep quality

o Self-reported depression and anxiety symptoms

o Physical activity

• It is good that the authors use Poisson regression with robust variance instead of logistic regression, given the outcome prevalence is common (>10 -15%). In describing the statistical analysis (model) used (for effect measure reporting), clearly justify your model choice, including why robust variance was used for technical clarity, and report the confidence interval (not mentioned).

o Suggested description: Given the high prevalence of the outcome, the association between GERD and sleep quality was evaluated using Poisson regression with robust variance to calculate crude prevalence ratios (cPR) and adjusted prevalence ratios (aPR) with 95% confidence intervals. Robust (sandwich) estimators were applied to account for potential overdispersion.

• The authors stated, “for inclusion of variables in the adjusted model, we followed an epidemiological approach by including all potential confounders of the association”.

o Authors should clearly state how confounders were selected. Mention whether clinical relevance (based on literature and biological plausibility, statistical criteria (variables with P < 0.2 in bivariate analysis), or a combination of both) was used to include the covariates within the model.

Results

• The authors stated, "Initially, 176 students were surveyed, of whom 5 did not meet the inclusion criteria; therefore, 171 participants were finally analyzed."

o Why were students who don’t meet the inclusion criteria surveyed in the first place? Rather, specific exclusion criteria should have been used to exclude patients.

o Modify the description accordingly.

• While doing descriptive statistics like in Table 1 (Characteristics of medical students from a private university in Lima, Peru), in addition to frequency values for all students, it would be better and more informative if participant characteristics were reported categorized for students with GERD and without GERD to account for group differences.

• The prevalence of GERD (76.6%) in your study is extremely high compared with previous studies, both in the general population and among medical students.

o What are the possible explanations? This should be clearly discussed in the discussion section.

Discussion and conclusion

• The alarmingly high prevalence of GERD compared with previous studies needs better discussion.

o Put the potential reasons for the previous studies’ higher, lower, and similar findings compared to your study separately.

• The authors stated that “Previous studies among Peruvian university students have reported similar or even lower prevalences of GERD,” referencing two studies. However, the statement is vague and inaccurate because, although the observed prevalence of GERD in the current study (76.6%) was comparable to that of Antenor Orrego Private University, Trujillo – Peru (75%), it was significantly higher than that of Huacho School of Human Medicine (42%).

o Given the significantly high GERD prevalence observed in the current study, the finding needs better comparison with those studies and contextualization.

• Moreover, the authors tried to forward potential reasons for the high prevalence of GERD, stating “high frequency of unhealthy dietary and behavioral habits identified in the sample, such as excessive consumption of coffee and energy drinks, use of NSAIDs, and intake of spicy foods, factors previously associated with an increased risk of GERD symptoms.”

o However, we don’t know whether the mentioned behaviors are more common in those students with or without GERD.

o As I mentioned before in the results section, reporting participant characteristics categorized for students with GERD and without GERD to account for group differences might be beneficial to give some perspective.

• While discussing the high prevalence of poor sleep, the authors stated: “This high frequency could be attributed to the intense academic workload, sustained stress, high prevalence of depression and anxiety, irregular schedules for studying and resting, and frequent use of electronic devices at night, factors that disrupt circadian rhythms and compromise both sleep duration and quality.”

o However, some of the factors mentioned (eg, depression and anxiety, frequent use of electronic devices at night) could have been measured in the study, which would also have mitigated the confounding factors in assessing the association between GERD and poor sleep.

• In the final paragraph, while discussing the association between GERD/GER symptoms with poor sleep, the authors have focused on describing findings from previous studies rather than discussing findings from the current study and analytically comparing it with previous reports. The whole paragraph needs to be modified accordingly.

Reviewer #3: The research question is timely and the use of adjusted Poisson regression is methodologically appropriate for this outcome. With revisions addressing the validity of the FSSG cut-off in this population, more transparent reporting of participation rates, inclusion of mental health covariates or explicit acknowledgment of their absence as a primary limitation, and more cautious interpretation of prevalence estimates and the dose–response magnitude, this work has the potential to make a meaningful contribution to the literature on student health in Latin America.

The cross-sectional design is appropriate for an initial exploration of this association, and the use of validated instruments (PSQI and FSSG) strengthens the methodological rigor. However, the reliance on convenience sampling and self-reported data introduces potential selection and reporting biases, which should be more explicitly acknowledged and discussed in terms of their impact on internal and external validity.

The FSSG cut-off of >8 was validated in a Japanese clinical population, not in Peruvian university students; authors must either cite a validated adaptation or reframe results as "GERD symptom burden" rather than clinical GERD prevalence.

A 1% prevalence increase per one-point FSSG increment is statistically significant but clinically modest; the authors should illustrate the magnitude by comparing predicted prevalence between a student at the GERD threshold (score 8) and one with moderate-severe burden (score ~30).

Reviewer #4: Refer to attached document for clarity.

Major comments for Authors – Action required)

1. Clarify and strengthen the knowledge gap (Abstract & Introduction)

The manuscript does not clearly articulate what is unknown in the specific context of Peruvian or Latin American medical students.

✔ Action required:

• Explicitly state the knowledge gap, e.g. lack of context-specific evidence in Peru

• Clearly define how this study adds to existing literature

2. Improve study setting description (Methods)

The study setting is described only as:

“a private university (Universidad Peruana Unión) in Lima, Peru”

This lacks sufficient contextual detail.

✔ Action required:

• Describe the institution (type, size, structure of medical program, student distribution)

3. Provide complete and reproducible sample size calculation (Methods)

The manuscript reports:

“minimum required sample size of 160 participants”

But omits:

• Allowable error (precision, e.g., 0.05)

• Formula used

• Finite population correction

• Non-response adjustment

✔ Action required:

• Provide full sample size formula and parameters

4. Address incomplete confounder adjustment (Critical Issue)

The manuscript includes lifestyle variables but omits:

• Mental health (stress, anxiety, depression)

• Medications affecting sleep

Yet later discusses:

“academic stress… depression and anxiety”

✔ Action required:

• Either:

o Include these variables in analysis (if available)

o OR strongly emphasize as a major limitation affecting validity

5. Correct misunderstanding of multicollinearity (Methods)

The manuscript states:

“age was categorized into tertiles to avoid multicollinearity”

✔ Action required:

• Revise this statement

• Provide correct approach to assessing multicollinearity (e.g., VIF)

6. Add data quality control procedures (Methods)

Data were collected via:

“Microsoft Forms… and WhatsApp”

But no mention of:

• Duplicate checks

• Missing data handling

• Validation procedures

✔ Action required:

• Describe data quality assurance measures

7. Improve statistical reporting in Results

Example:

“prevalence was significantly higher… (p=0.016)”

✔ Issues:

• Over-reliance on p-values

• No effect size in Table 2

✔ Action required:

• Include crude prevalence ratios (cPR) with 95% CI in Table 2

• Reduce emphasis on p-values alone

8. Improve Table clarity and completeness

• Table 2 lacks effect size

• Legend does not specify which test was used for each variable

• Table 1 does not clearly separate counts and percentages

✔ Action required:

• Add cPR and CI

• Clarify statistical tests per variable

• Improve formatting

9. Remove interpretation from Results section

Example:

“a linear and directly proportional relationship was observed…”

✔ Action required:

• Replace with neutral wording (e.g., “a positive association was observed”)

10. Avoid overinterpretation in Discussion

Example:

“could be explained by unhealthy dietary and behavioral habits…”

✔ Issue:

• Implies causal mechanisms

✔ Action required:

• Use cautious language (“may be associated with”)

11. Clearly state study contribution (Discussion)

The manuscript compares findings but does not explicitly state:

• What this study adds

✔ Action required:

• Add a clear statement of contribution

12. Separate implications and recommendations

The section:

“Implications and recommendations”

mixes interpretation and action.

✔ Action required:

• Separate into:

o Implications

o Recommendations

13. Strengthen limitations section

The current section does not fully reflect:

• Selection bias due to recruitment method

• Online survey limitations

• Lack of local validation of FSSG

• Residual confounding

✔ Action required:

• Expand limitations to explicitly include:

o Recruitment bias (health-conscious participants)

o Online survey bias

o Measurement limitations

o Missing confounders

14. Major Comments on Limitations (For Authors – Action Required)

1. Selection bias not sufficiently specified

The manuscript states:

“Convenience sampling may limit external validity and generate selection bias.”

This is too general and does not reflect the actual recruitment approach:

• In-person academic/health-related event

• Online dissemination via WhatsApp

Issue:

• Likely overrepresentation of:

o More engaged

o More health-conscious students

This may bias:

• GERD prevalence (76.6%)

• Poor sleep quality (84.8%)

✔ Action required:

• Explicitly describe how recruitment method may have introduced selection bias

2. Online survey limitations not explicitly acknowledged

Although reporting bias is mentioned:

“students experiencing these problems may have been more likely to respond…”

The manuscript does not explicitly state that:

• Data were collected via an online self-administered questionnaire

Missing:

• Self-report bias

• Recall bias

• Lack of supervision

• Potential duplicate or careless responses

✔ Action required:

• Add a clear limitation related to online data collection and data quality

3. Residual confounding underemphasized (critical issue)

The manuscript states:

“variables such as academic stress or other mental health variables were not measured…”

This is appropriate but understated

Issue:

• Mental health factors:

o Strongly affect both GERD and sleep quality

• Their omission is a major threat to internal validity

✔ Action required:

• Emphasize this as a key limitation that may bias results

4. Omission of medications affecting sleep

The study includes:

• NSAIDs

But does not mention omission of:

• Sleep medications

• Antidepressants

• Stimulants

Issue:

• These are important confounders influencing sleep outcomes

✔ Action required:

• Add this as a limitation contributing to residual confounding

5. Lack of local validation of measurement tools

The manuscript states:

“validated and widely used instruments were employed…”

However:

• PSQI → validated locally ✔

• FSSG → not clearly validated in Peruvian population

Issue:

• Possible measurement bias or cultural misclassification

✔ Action required:

• Acknowledge limited local validation of FSSG

6. Reporting bias explanation needs strengthening

The manuscript states:

“students experiencing these problems may have been more likely to respond…”

This is valid but incomplete

✔ Action required:

• Link explicitly to:

o Voluntary participation

o Online format

o Recruitment strategy

14. Make recommendations actionable (Conclusion)

Current statement:

“implementing systematic screening… and developing timely interventions…”

✔ Issue:

• Too vague

✔ Action required:

• Specify:

o Screening tools (PSQI, GERD tools)

o Lifestyle interventions (diet, caffeine reduction)

o Clinical pathways (student health services)

Minor Comments (For Authors – Action Required)

1. Title refinement

• Reduce redundancy (“association between” + “cross-sectional study”)

2. Keywords

• Remove study design

• Focus on content terms (GERD, sleep quality, students, Peru)

3. Abstract

• Add:

o Knowledge gap

o Contribution

o Effect estimates (aPR with CI)

4. Reporting of missing data

• No mention of:

o Missing values

o Handling approach

✔ Add this to Methods/Results

5. High prevalence values

• GERD (76.6%) and poor sleep (84.8%) are very high

✔ Briefly acknowledge possible reasons (selection bias, measurement)

6. Strengths section

• Expand to include:

o Appropriate analytical approach (PR instead of OR)

o Use of standardized tools

7. Minor Comments on Limitations

1. Screening tools vs diagnosis

The manuscript states:

“FSSG and PSQI are screening instruments…”

✔ Appropriate

Could be improved by adding:

• Risk of misclassification bias

2. Balance of strengths vs limitations

• Limitations are present but:

o Not sufficiently detailed

• Strengths somewhat overstated relative to limitations

✔ Suggest improving balance

6. PLOS authors have the option to publish the peer review history of their article (what does this mean?). If published, this will include your full peer review and any attached files.

Reviewer #1: No

Reviewer #2: No

Reviewer #3: No

Reviewer #4: **Yes:** Evelyn Kigenyi Zalwango

---

## [Author Response · Author response to Decision Letter 1]

10 Apr 2026

RESPONSE TO REVIEWERS

Manuscript title: Association between gastroesophageal reflux symptoms and sleep quality among medical students at a private university in Lima, Peru: a cross-sectional study

We sincerely thank the Academic Editor and all four Reviewers for their thorough, constructive, and detailed evaluation of our manuscript. Their comments have led to substantial improvements in the clarity, methodological transparency, and scientific rigor of the revised version. We have addressed every comment carefully, point by point, in the sections below. Changes made to the manuscript are described in each response, and the specific location of each change is noted.

Reviewer 1

Comment 1.1."In the methods section, consider presenting the sample size calculation first, then the recruitment approach, and finally how incomplete records were handled. The study does not report the number of students approached or the number who declined participation."

We thank the reviewer for this organizational suggestion. We have reorganized the "Design, population, and sample" subsection so that the sample size calculation is presented first, followed by the sampling and recruitment approach, and finally the handling of incomplete data. However, due to convenience sampling we do not have the number of students invited to participate.

Modified text in Methods: “A sample size calculation was performed considering a population of 698 students, a proportion of poor sleep quality of 83.9% [17], and a 95% confidence level, resulting in a minimum required sample size of 160 participants. Non-probabilistic convenience sampling was used.”

Comment 1.2. "The definition of poor sleep quality requires clarification. The manuscript classifies poor sleep as a PSQI score ≥5; however, the validated cutoff defines poor sleep quality as a global PSQI score >5, with scores ≤5 indicating good sleep quality. The authors should clarify/justify the use of this alternative threshold."

We thank Reviewer 1 for this important methodological observation. The original validation study by Buysse et al. (1989) mentions “A post hoc cutoff score of 5 correctly identified 88.5% (131/ 148) of all patients and controls (kappa = 0.75, p < 0.001)”. Also, in two Peruvian validation studies (Ravelo Bobadilla, 2022; and Luna-Solis, 2015) a cutoff of ≥5 was applied. We have added these references.

Comment 1.3. "The link between multicollinearity assessment (VIF) and the decision to categorize age into tertiles requires clarification. VIF is used to detect collinearity among predictors. Please clarify the methodological basis for this decision."

We appreciate this observation. We acknowledge that the original wording was imprecise.

Modified text in Methods: “Multicollinearity was assessed using the variance inflation factor (VIF). We observed that age, when included as a continuous variable, showed a high degree of multicollinearity. To reduce this collinearity while retaining age as a covariate in the adjusted model, age was categorized into tertiles.”

Comment 1.4. "Given the findings of reference 14, the incremental contribution of this study is not immediately clear. The discussion should explicitly articulate how this study advances existing knowledge, including any novel population-specific insights or methodological contributions."

We thank reviewer for this important point. We have added a clear statement of the study's incremental contributions to the Discussion section.

Added paragraph in the Discussion section: “Our study provides several incremental contributions to the existing literature on the relationship between GERD and sleep quality. First, it focuses on Peruvian medical students, a specific and understudied subgroup within the Latin American context. Second, unlike previous meta-analysis [14], which was based primarily on studies from the general population, our study analyzes a population with a distinct risk profile characterized by high academic stress, irregular dietary habits, and young age. Third, by using Poisson regression with robust variance, we obtained adjusted prevalence ratios that are more appropriate for cross-sectional, representing a methodological improvement over previous studies in this population that relied mainly on unadjusted comparisons. Finally, we identified a dose–response relationship between GERD symptom burden and poor sleep quality, which strengthens the biological plausibility of the observed association in this specific population.”

Reviewer 2

Comment 2.1. "Overall, check for grammatical flow and sentence formation of the entire manuscript, as some of the sentences are long, vague, and not up to the scientific standard."

We thank Reviewer 2 for this comment. We have carefully revised the entire manuscript for grammatical clarity, sentence structure, and scientific precision.

Comment 2.2. "Limit abbreviations in the abstract section."

We thank the Reviewer for this observation. We have revised the Abstract to reduce the use of abbreviations. Specifically, we removed the abbreviations for the Pittsburgh Sleep Quality Index and the Frequency Scale for the Symptoms of Gastroesophageal Reflux Disease. We retained GERD because it is used repeatedly throughout the Abstract.

Comment 2.3 (Introduction). "The authors didn't clearly describe what is already known regarding the association (and its degree) between sleep quality and GERD symptoms, particularly in the study population (medical students). Which gaps in the literature does the current study aim to narrow?"

We appreciate this important observation. We have expanded the Introduction to more explicitly describe: (1) what is known about the GERD–sleep quality association from the systematic review by Tan et al.; Reference 14; (2) what is known from the two prior cross-sectional studies in medical students (Teimouri & Amra, 2021; Sharma et al., 2018; References 15 and 16), highlighting their limitations — specifically, the absence of adjusted analyses; and (3) a clear statement of the knowledge gap: no study with adequate control for confounders has been conducted among Peruvian medical students, and context-specific evidence from this population is needed to guide local health interventions.

Added text in Introduction: “A systematic review found that GERD is significantly associated with poorer sleep quality (OR: 1.47; 95% CI: 1.24 to 1.74) and shorter sleep duration (OR: 1.17; 95% CI: 1.12 to 1.21), although with some heterogeneity among studies [14]. Evidence in medical students is still scarce. A study in Iran reported an association between reflux symptoms and impaired sleep quality, while a study in India found that inadequate sleep and sleep-related habits were associated with GERD symptoms; however, both studies were cross-sectional and lacked adequate adjusted analyses to control for confounding [15,16]. This is particularly important in medical students, a population with a high frequency of poor sleep quality, irregular eating habits, and heavy academic workload, factors that may modify this relationship. To date, no study with adequate control for confounding has evaluated this association among Peruvian medical students. Therefore, context-specific evidence is needed to better characterize this relationship and support local preventive strategies.”

Comment 2.4 (Methods) "The study period should be stated (e.g., from October 27 to November 28, 2024)."

We thank Reviewer for this comment. The study period (October 27 to November 28, 2024) was already stated in the Procedures subsection of the Methods. To improve visibility and completeness, we have also added the study period to the first paragraph of the Methods section ("Design, population, and sample").

Modified text in Methods: “An analytical cross-sectional study was conducted among medical students from a private university (Universidad Peruana Unión) in Lima, Peru, between October 27 and November 28, 2024.”

Comment 2.5 (Methods). "Inclusion and exclusion criteria should be clearly delineated and justified. Given that the study population is medical students (who are most likely more than or somewhere around 18 years old), what is the rationale for using the 18-year-old age cutoff for inclusion? Were the exclusion criteria limited to participants with incomplete data? Please mention if any additional exclusion was used. What defines incomplete data in the study?"

We thank Reviewer for these important questions. We have revised the Methods section to clearly separate and describe the inclusion and exclusion criteria. The inclusion criteria were medical students enrolled at Universidad Peruana Unión, aged ≥18 years, who voluntarily provided informed consent to participate. The exclusion criterion was the presence of at least one missing value in the study variables collected through the survey; however, no participants were excluded for missing data in the final dataset. Regarding the minimum age requirement of 18 years, although most medical students are adults, some first-year students may be younger than 18 years. This cutoff was established to ensure that all participants had full legal capacity to provide informed consent under Peruvian regulations. No additional exclusion criteria were applied.

Comment 2.6 (Methods) "In participant selection, why do you prefer to include only students attending an in-person academic event? This can introduce a potential selection bias because students with frequent GERD symptoms and poor sleep may have increased absenteeism, which could lead to their exclusion and underrepresentation in the study."

We acknowledge that recruiting participants during an in-person academic event could raise concerns about potential selection bias if students with more severe GERD symptoms or poorer sleep quality were less likely to attend. However, this event is held annually by the medical school, includes students from all academic years, and is usually mandatory as part of academic activities contributing to course assessment. Therefore, the likelihood of substantial underrepresentation due to absenteeism is likely low. In addition, to broaden participation beyond those present at the event, we also disseminated the survey link via WhatsApp during the data collection period.

Modified text in Methods, Procedures: “The survey was distributed during an in-person academic event organized by the medical school, which included students from all academic years. This event is usually mandatory and forms part of the academic evaluation, which likely reduced the possibility of substantial selection bias due to absenteeism.”

Comment 2.7 (Methods). "Why did you prefer to use non-probabilistic convenience sampling? Even though non-probabilistic convenience sampling can be useful for pilot studies or exploratory research, in analytical studies like this, it can pose a significant limitation. The authors should clarify these points and acknowledge them as limitations if necessary."

We appreciate this comment. Non-probabilistic convenience sampling was adopted due to logistical constraints, as no official list of enrolled students with contact information was available to enable probability-based sampling. We acknowledge that this approach limits the external generalizability of our findings and may introduce selection bias. We have added a brief justification of this methodological choice in the Methods section.

Modified text in Methods: “Non-probabilistic convenience sampling was used because no official student list was available to perform random sampling.”

Comment 2.8 (Methods). "Was block sampling by academic year used to affirm the balanced distribution of participants across academic years?"

We thank the reviewer for this question. Block sampling by academic year was not used. For the same reason described above, we did not have access to an official student list that would allow random selection within academic-year strata. Therefore, non-probabilistic convenience sampling was used. Nevertheless, the survey was distributed during an academic event that included students from all academic years, which facilitated the participation of students across the different stages of training.

Comment 2.9 (Methods). "Was sample size calculation for detecting associations performed?"

We thank Reviewer for raising this point. A formal statistical power calculation for detecting a specific magnitude of association was not performed, as no prior reliable estimate of the association between GERD and sleep quality in this specific population was available at the time of study planning. We have added this clarification to the Methods section.

Modified text in Methods: “A sample size calculation specifically for detecting an association was not performed because no preliminary evidence was available in this population using the same instruments to assess both GERD symptoms and sleep quality. Therefore, the sample size was estimated using a descriptive approach, considering a population of 698 students, a proportion of poor sleep quality of 83.9% [17], and a 95% confidence level, resulting in a minimum required sample size of 160 participants.”

Comment 2.10 (Methods). "The dependent variable in the study was sleep quality, specifically "during the last month". Do you use a similar time frame for the independent variable (GERD symptoms)? This is critical to ensure a temporal association between GERD symptoms and sleep quality."

We thank Reviewer 2 for this critical observation. The PSQI assesses sleep quality over the past month. The FSSG, as originally developed by Kusano et al. (2004; Reference 19), does not specify an explicit recall period; response options reflect habitual symptom frequency ("never," "occasionally," "sometimes," "often," "always") without a defined time window. We acknowledge that this difference in temporal framing between the two instruments is a methodological limitation, as the GERD symptom burden captured by the FSSG may not correspond precisely to the same one-month period assessed by the PSQI. We have added this as an explicit limitation in the revised manuscript.

Added text in Discussion, Limitations: “Furthermore, the PSQI assesses sleep quality during the last month, whereas the FSSG does not specify an explicit recall period, which may have introduced temporal misalignment between the measurement of the exposure and the outcome.”

Comment 2.11 (Methods). "Some relevant confounders based on prior literature seem to be missing. This includes use of symptomatic treatments for GERD, like PPIs; comorbidities, including psychiatric illness and other medical conditions known to impact sleep quality; self-reported depression and anxiety symptoms; physical activity."

We acknowledge that some potentially relevant variables, such as physical activity and mental health-related factors, may act as confounders in the association between GERD symptoms and sleep quality and were not measured in this study. In contrast, the use of medication for GERD was not considered a confounder, since it is more likely to affect GERD symptom burden directly rather than sleep quality independently. Nevertheless, we recognize that the absence of some unmeasured variables may have introduced residual confounding, and this has been explicitly acknowledged in the Limitations section: “Another important factor is that variables such as academic stress or other mental health variables were not measured, which could have resulted in residual confounding.”

Comment 2.12 (Methods). "In describing the statistical analysis, clearly justify your model choice, including why robust variance was used for technical clarity, and report the confidence interval."

We thank the reviewer for this suggestion. We have revised the Statistical Analysis paragraph to more clearly justify the use of Poisson regression with robust variance.

Added text in Methods: “This regression approach was used given the binary nature of the outcome. Robust (sandwich) variance estimators were applied to obtain valid standard errors when Poisson models are fitted to binary data. “

Comment 2.13 (Methods) "Authors should clearly state how confounders were selected. Mention whether clinical relevance, statistical criteria, or a combination of both was used."

W

---

## [Decision Letter · Decision Letter 1]

23 Apr 2026

Gastroesophageal reflux symptoms and sleep quality among medical students at a private university in Lima, Peru: a cross-sectional study

PONE-D-26-11670R1

Dear Dr. Soriano-Moreno,

We’re pleased to inform you that your manuscript has been judged scientifically suitable for publication and will be formally accepted for publication once it meets all outstanding technical requirements.

Kind regards,

Taeyun Kim

Academic Editor

PLOS One

Additional Editor Comments (optional):

All comments have been adequately addressed in this revision.

Reviewers' comments:

Reviewer's Responses to Questions

**Comments to the Author**

1. If the authors have adequately addressed your comments raised in a previous round of review and you feel that this manuscript is now acceptable for publication, you may indicate that here to bypass the “Comments to the Author” section, enter your conflict of interest statement in the “Confidential to Editor” section, and submit your "Accept" recommendation.

Reviewer #2: All comments have been addressed

Reviewer #4: All comments have been addressed

2. Is the manuscript technically sound, and do the data support the conclusions?

Reviewer #2: Yes

Reviewer #4: Yes

3. Has the statistical analysis been performed appropriately and rigorously? 

Reviewer #2: Yes

Reviewer #4: Yes

4. Have the authors made all data underlying the findings in their manuscript fully available?

Reviewer #2: Yes

Reviewer #4: Yes

5. Is the manuscript presented in an intelligible fashion and written in standard English?

Reviewer #2: Yes

Reviewer #4: Yes

6. Review Comments to the Author

Reviewer #2: The authors have done a remarkable work in revising the manuscript based on the reviewers comments. I believe the manuscript is now ready for acceptance to publication after proofreading for some minor issues.

Some minor issues:

- Defining abbreviation on first encounter: eg. TRPV1 (Transient Receptor Potential Vanilloid 1) in introduction section.

- In the Introduction section: Their is no need of stating the CI for the results from the referenced studies. Just mentioning the degree of the observed association and whether it was significant association is enough. This will improve the conciseness and readability.

- In the conclusion section: "These findings highlight the use of simple screening tools for GERD symptoms and sleep quality in university health settings and the implementation of targeted interventions focused on sleep hygiene and reduction of reflux-related behaviors." The sentence is imprecise and seems to overstate conclusions.

The phrase "highlight the use of" could be misleading here as your study doesn't assess the validity or use of screening tools and the use of implementing targeted interventions. Rather use alternative phrases like "highlight the need for", "Emphasize the need for".

Carefully search for similar wording problems which may mislead readers from capturing what you are really trying to express.

Reviewer #4: Thank you for your thorough and thoughtful responses to the reviewer comments. You have addressed all concerns satisfactorily, and the manuscript has improved significantly in clarity, coherence, quality, and scientific contribution. Well done.

7. PLOS authors have the option to publish the peer review history of their article (what does this mean?). If published, this will include your full peer review and any attached files.

Reviewer #2: No

Reviewer #4: **Yes:** Eva Kigenyi Zalwango

---

## [Editor Report · Acceptance letter]

PONE-D-26-11670R1

PLOS One

Dear Dr. Soriano-Moreno,

I'm pleased to inform you that your manuscript has been deemed suitable for publication in PLOS One. Congratulations! Your manuscript is now being handed over to our production team.

Kind regards,

on behalf of

Dr. Taeyun Kim

Academic Editor

PLOS One